Cloning and comparative modeling identifies a highly stress tolerant Cu/Zn cytosolic super oxide dismutase 2 from a drought tolerant maize inbred line

Gautam Anuradha 1
Khan Fatima Nazish 2 3
Priya Surabhi 1
Kumar Krishan 1
Sharda Shivani 4
Kaul Tanushri 5
Singh Ishwar 1
Langyan Sapna 6 singh.sapna06@gmail.com
Yadava Pranjal 1 2 pranjal@iari.res.in
1 ICAR-Indian Institute of Maize Research, Pusa Campus , New Delhi , India
2 Division of Plant Physiology, Indian Agricultural Research Institute, Pusa , New Delhi , India
3 Department of Biotechnology, Jamia Millia Islamia, Jamia Nagar , New Delhi , India
4 Amity Institute of Biotechnology, Amity University , Noida , India
5 International Centre for Genetic Engineering and Biotechnology, Aruna Asaf Ali Marg , New Delhi , India
6 Division of Germplasm Evaluation, ICAR-National Bureau of Plant Genetic Resources, Pusa Campus , New Delhi , India
Batra Lalit
Electronic publication date: 2023 Mar 13
Publication date: 2023
Volume: 11
Electronic Location ID: e14845
Received 2022 Sep 9; Accepted 2023 Jan 11
Copyright: © 2023 Gautam et al.
Copyright year: 2023
Copyright holder: Gautam et al.
License: This is an open access article distributed under the terms of the Creative Commons Attribution License, which permits unrestricted use, distribution, reproduction and adaptation in any medium and for any purpose provided that it is properly attributed. For attribution, the original author(s), title, publication source (PeerJ) and either DOI or URL of the article must be cited.
License URL: https://creativecommons.org/licenses/by/4.0/

Keywords: Zea mays L., Superoxide dismutase, SOD2, Drought stress, Bioinformatic analysis

Funding: National Agricultural Science Fund of Indian Council of Agricultural Research (NASF/GTR-5004/2015-16/204) Junior Research Fellowship from Core Research Grant Project of Science and Engineering Research Board CRG/2019/005451 Senior Research Fellowship from ICAR NPTC 3015 This work was supported by a research grant from the National Agricultural Science Fund of Indian Council of Agricultural Research (ICAR) (NASF/GTR-5004/2015-16/204) to Pranjal Yadava and Tanushri Kaul. Fatima Nazish Khan is supported by the Junior Research Fellowship from Core Research Grant project of the Science and Engineering Research Board (CRG/2019/005451) to Pranjal Yadava. Anuradha Gautam was supported by a Senior Research Fellowship from ICAR (NPTC 3015). The funders had no role in study design, data collection and analysis, decision to publish, or preparation of the manuscript.

==============================
Plants have a complex system of stress response that deals with different types of stresses. Maize (Zea mays L.), one of the most important crops grown throughout the world, across a range of agro-ecological environments, employs complex mechanisms of gene regulation in response to drought stress. HKI 335 is a tropical maize inbred line showing remarkable adaptation to drought stress. Abiotic stresses, like drought, trigger the production of reactive oxygen species (ROS) due to the incomplete reduction or excitation of molecular oxygen, eventually leading to cell damage. Superoxide dismutase (SOD, EC 1.15.1.1) is a metalloenzyme that acts as the first line of defense against ROS. We cloned the Sod2 gene from HKI 335 inbred line and analyzed its protein through detailed in silico characterization. Our comparative modeling revealed that at the level of tertiary structure, the HKI 335 SOD2 protein is highly similar to Potentilla atrosanguinea SOD2, which had been previously identified as highly thermostable SOD that can tolerate autoclaving as well as sub-zero temperatures. We performed phylogenetic analysis, estimated physicochemical properties, post-translational modifications, protein-protein interactions, and domain composition of this SOD2. The phylogenetic analysis showed that orthologous sequences of SOD from different species were clustered into two clusters. Secondary structure prediction indicates that SOD2 is a soluble protein and no transmembrane domains have been found. Most of the beta sheets have RSA value greater than 2. The Ramachandran plot from PDBsum revealed that most of the residues fall in the highly favored region. It was estimated that the value of the instability index was less than 40, the value of the aliphatic index was extremely high and the GRAVY value lies between −2 and +2. We could identify only one phosphorylation site, located at position 20 with a score of 0.692. Overall, the unique stress-tolerant properties of the HKI 335 SOD2, may be one of the reasons contributing to the high drought tolerance trait exhibited by HKI 335 maize inbred line. Further research may reveal more insights into the drought adaptation mechanism in maize and the eventual deployment of the trait in maize hybrids.

Introduction

To meet the increasing demand due to rising population and economic development, the production of crops should maintain a sustainable growth (Brown & Funk, 2008; Liu et al., 2019). However, environmental stresses such as drought, have become a major limitation in crop production, under unpredictable weather and changing climate patterns (Takeda & Matsuoka, 2008). Maize (Zea mays L.), is the most produced and consumed grain of the world, with diverse uses as food, feed, fuel and industrial products (Schnable et al., 2009). It is grown worldwide across a range of agro-ecological environments, and is therefore prone to abiotic stresses, like drought. Although, the productivity of maize worldwide is high, yet the crop is highly sensitive to drought (Lee et al., 2020; Rodrigues et al., 2021). Drought has been an ecological crisis throughout the world and is one of the major stresses that limits crop productivity (Bartels & Sunkar, 2005). It is also considered to be the major serious environmental factor that limits the productivity of maize, especially under rain-fed conditions (Rao et al., 2016). The maize crop is susceptible to drought at different growth stages such as grain-filling, pre-flowering and seedling (Liang et al., 2020). Therefore, it is essential to understand the mechanism of drought-tolerance and breed drought-tolerant varieties (Ferdous, Hussain & Shi, 2015). When plants are exposed to water stress, an integrated and diverse stress response system is activated that consists of various adaptive mechanisms at all scales of organization: morphological, physiological, and molecular. During the early stages of drought, plants absorb underground water efficiently, partially close stomata, check transpirational water loss, and finetune the metabolic rate to match with the available carbon and other resources. Osmolytes such as prolines, soluble sugars, spermines, and glycine betaine are necessary to retain the cell turgor pressure, under stress. Antioxidant enzymes such superoxide dismutase (SOD), ascorbate peroxidase (APX), catalase (CAT), and glutathione reductase (GR) are critical to maintain survival redox state for the cells under stress. Differential gene and protein expression through transcriptional and other regulation is hallmark of drought stress adaptation response (Wang et al., 2016). Abiotic stress such as drought, extreme temperatures, high salinity, excessive light, pollutants such as ozone and herbicides, high concentrations of heavy metals, excessive UV radiation, and others (Lee et al., 2020) trigger the production of reactive oxygen species (ROS) due to incomplete reduction (hydrogen peroxide –H2O2; superoxide radical –O•−2; hydroxyl radical –HO•, etc.) or excitation (singlet oxygen –1O2) of molecular oxygen, eventually leading to cell damage (Rodrigues et al., 2021).

Superoxide dismutase (SOD, EC 1.15.1.1) is a metalloenzyme that acts as the first line of defense against ROS. SODs play a major role in detoxification of ROS. SODs catalyze the dismutation of the superoxide radical (O2−) into oxygen and hydrogen peroxide (H2O2). All aerobic organisms that are prone to oxidative stress require SODs to dismutase O2 to yield H2O2. SODs are categorized in accordance with their metal cofactors and subcellular distribution: mitochondrial manganese SOD (MnSOD), cytosolic copper/zinc SOD (Cu/ZnSOD), and iron SOD (FeSOD) (Lee et al., 2020). Maize contains nine SOD isozymes viz. four Cu/Zn cytosolic isozymes (SOD2, SOD4, SOD4A and SOD5) four mitochondrial associated MnSODs (SOD3.1, SOD3.2, SOD3.3 and SOD3.4) and one Cu/Zn chloroplast-associated isozyme (SOD-1). There are many reports stating that SOD2 overexpressing transgenic plants show enhanced tolerance to oxidative stresses such as drought.

A hyper-thermostable version of SOD2 enzyme, has been isolated from Potentilla atrosanguinea which was further engineered by mutating single amino acid to increases its thermal stability by two fold. P. atrosanguinea is reported to be functional at sub-zero temperature to >50 °C and also the protein retain activity after autoclaving (heating at 121 °C, at a pressure of 1.1 kg per square cm for 20 min). Cu, Zn SOD from P. atrosanguinea provides a unique opportunity to understand the role of free Cys-95. This enzyme is a dimer consisting of two asymmetric subunits, namely A and B. Each subunit consists of eight stranded β-sandwich (1a–8h) connected by seven loops (I–VII), one short helix and one 310 helix. As increased thermostability is desirable for Cu, Zn SOD, to act as a potent antioxidant enzyme (Walia et al., 2018). The expression of the SOD genes from plants is influenced by plant growth stage and growth regulating substances. Abscisic acid and osmotic stress induce expression of Cu/ZnSOD and MnSOD genes in maize. The overall SOD expression is determined at transcriptional, posttranscriptional, and translation levels.

In our earlier screens, HKI 335 maize inbred line (derived from Pool 10, All India Coordinated Research Project on Maize, Karnal centre, India) was found to be a highly drought tolerant line (Nepolean et al., 2013; Yadava, Kaur & Singh, 2013; Yadava, 2016). Previously, we reported that enhanced SOD activity in maize could be a key strategy employed by the plant to mitigate abiotic stress, as evident by transcriptional modulation of antioxidant genes during methyl viologen induced oxidative stress (Kumar, Sahoo & Ahuja, 2002; Lee et al., 2009, 2010; Kumar et al., 2012; Lobell et al., 2014; Lee et al., 2016; Singroha, Sharma & Sunkur, 2021). This prompted us to clone and analyze Cu/Zn cytosolic Sod2 from the highly drought tolerant line HKI 335.

Materials and Methods

RNA isolation

Leaf tissue was sampled from 21-day old maize seedlings (Zea mays, HKI 335 inbred line) growing in pot under normal controlled conditions in greenhouse. and RNA was isolated using RNA isolation kit (Genetix Life Sciences, New Delhi, India). Quality and concentration of RNA was checked on Nanodrop Spectrophotometer (Thermo Fisher, Waltham, MA, USA). 1 µg of RNA was used for cDNA synthesis using oligodT primers (cDNA synthesis kit; Invitrogen, Waltham, MA, USA).

PCR amplification

Gene specific primers were designed and synthesized for maize (Sod2) (sod2_Forward ACAATGGTGAAGGCAGTTGCTGTC and sod2_Reverse TTAGCCTTGGAGCCCAATGATACC) and amplification was done by polymerase chain reaction (PCR) using high fidelity polymerase (Pfu). The amplified product was ligated into pJET vector and transformed into competent E. coli cells (DH5α strain). Colony PCR was done using above primers to identify transformed colonies and plasmid was isolated. This plasmid was sent for sequencing. BLAST analysis of sequencing results confirmed our sequences as SOD2 mRNA which were then submitted in NCBI (ALF00121.1). The sequences were retrieved for further bioinformatic analysis.

Sequence search

The target sequence was searched for similar sequence using the BLAST (Basic Local Alignment Search Tool) (https://blast.ncbi.nlm.nih.gov/Blast.cgi?PAGE=Proteins) against Protein Database (PDB) (Johnson et al., 2008). Only full-length protein sequences were considered for in silico analysis.

Multiple sequence alignment and phylogenetic analysis

Similar sequences of SOD2 were searched using BLAST and multiple sequence alignment was done using Clustal W (https://www.genome.jp/tools-bin/clustalw), in order to infer the phylogenetic relationship between members of the antioxidant gene family (Thompson, Gibson & Higgins, 2003). The final alignment was completed through manual editing. The FASTA multiple sequence alignment was used to infer the neighbor-joining (NJ) phylogenetic tree using the MEGA 6.0 software (Tamura et al., 2013). The bootstrap interior branch test, superior to bootstrap analysis in testing confidence in a given tree node, was applied to test the statistical significance of the SOD2 of NJ phylogenetic tree.

Secondary structure prediction

The target sequence in FASTA format have been used for prediction of secondary structure of protein. With the utilization of online bioinformatics tool known as SABLE (http://sable.cchmc.org/), one can identify the secondary structure of target protein sequence (Adamczak, Porollo & Meller, 2005). This tool generally predicts real valued relative Solvent AccessiBiLitiEs of amino acid residues in protein. It uses evolutionary profiles for improved prediction of secondary structure. The result of secondary structure was visualized using POLYVIEW-2D.

Modelling of protein’s three-dimensional (3D) structure

The 3D structure of the protein was identified using online bioinformatics tool called MODWEB (https://modbase.compbio.ucsf.edu/modweb/) (Eswar et al., 2003). It modeled the structure using comparative protein structure modeling, which can calculate comparative models for a large number of protein sequences, using many different template structures and sequence-structure alignments.

Model validation

The model was evaluated on the basis of geometrical and stereo-chemical constraints using PDBsum (http://www.ebi.ac.uk/pdbsum) (Laskowski et al., 2018) and ProSA-Web (https://prosa.services.came.sbg.ac.at/) (Wiederstein & Sippl, 2007; Gill & Tuteja, 2010). PDBsum is used for validation of PDB structure through which stereochemical quality and accuracy of the predicted model is evaluated, which is based on Ramachandran plot calculation. Amino acid residues in proteins were examined by torsion angles Ø and Ψ and a percentage quality measurement of the protein structure was used, in which four sorts of occupancy were called core, allowed, generously allowed and disallowed regions. SuperPose (http://superpose.wishartlab.com/) was used for protein superposition that calculates the superpositions using a modified quaternion approach (Maiti et al., 2004). We calculated the root mean squared deviation (RMSD) to identify the distance between two objects by superposing the two structures and is performed between equivalent atom pairs.

Physiochemical properties

The physiochemical properties of the target protein such as molecular weight, number of amino acids, theoretical isoelectric point (pI), total number of negatively charged (Asp + Glu) and positively charged (Arg + Lys) residues, amino acid and atomic composition, extinction coefficient, estimated half-life, grand average of hydropathicity (GRAVY), instability and aliphatic index are identified using online tool ExPASy’s ProtParam (http://expasy.org/tools/protparam.html) (Gasteiger et al., 2005).

PTM site prediction

For the prediction of PTM sites, the online bioinformatics tool PTM-ssMP (Liu et al., 2018) was explored.

Protein-protein interaction analysis

For predicting the function of query sequence with the reference sequence, protein-protein interaction (PPI) networks analysis has been done using online tool STRING (http://string-db.org/) (Szklarczyk et al., 2019).

Domain and motif identification

We identified the domain composition of target protein sequence by using the bioinformatics tool SMART (http://smart.embl-heidelberg.de/) (Letunic, Doerks & Bork, 2012). It annotates and identifies the genetically mobile domains and analyses the domain structure. It can also detect extracellular, signalling and chromatin associated proteins bearing more than 500 domain families. ScanProsite server (https://prosite.expasy.org/scanprosite/) have been used for identification of signature motif in the query protein sequence (De Castro et al., 2006).

Results

Cloning of Sod2 from HKI 335 maize inbred line

The ZmSod2 gene was amplified from cDNA using gene specific primers and cloned in standard plasmid vector, followed by sequencing the ZmSod2 HKI 335 sequence was deposited in the GenBank database of National Centre for Biotechnology Information (NCBI), USA with accession number ALF00121.1. The cloned sequence (151 amino acid) contained all the characteristic conserved domains of Cu-Zn Superoxide Dismutase superfamily (PLN02386 family protein) (Fig. S1).

Phylogenetic analysis

The multiple sequence alignment shows aligning of SOD2 from different stress tolerant plant species. Similar sequences of SOD2 from different species were aligned and further used for phylogenetic analysis using the neighbour joining (NJ) method. The results show that SOD2 in different plant species demarcated into two prominent clusters i.e., monocots (6) and dicots (5) (Fig. 1).

Figure 1 (A) Multiple sequences alignment using Clustal W from different plant species for SOD2 sequences.

Blue color letters indicate that there is substitution of amino acid at that particular position, (B) Phylogenetic analysis showing Zea mays SOD2 as compared with SOD2 proteins from other plant species.

Secondary structure and comparative modeling

Using an online bioinformatics tool known as SABLE, we predicted the secondary structure of the target protein sequence (Fig. 2). The secondary structure of the protein sequence viewed by polyview-2D show the absence of alpha-helices. At the same time, there are a large number of coils and beta-sheets with fully exposed relative solvent accessibility (RSA). The result indicate that it is a soluble protein and no transmembrane domains have been found (Fig. S2). Most of the beta-sheets have RSA value greater than 2, which shows that the amino acids in the protein secondary structure are completely buried. The high value of the RSA of the target protein suggests that the protein is stable for a number of biological functions within the cell. The theoretical structure of SOD2 was generated using MOD web by comparative modeling of protein structure prediction through which identical or non-identical information about the target sequences is analyzed. We developed 14 models calculated for SOD2 out of which two models were selected. The model with LONGEST_DOPE had a score of −1.834 by using Modweb which was dynamically refined and validated. Comparative modeling revealed that the cloned SOD2 exhibits 83% identity with Potentilla atrosanguinea SOD2 (PDB ID: 2Q2L_A) with an e-value of 4e−86 (Figs. 3A and 3B).

Figure 2 Secondary structure predictions for ZmSOD2 protein from drought tolerant inbred HKI 335.

The colour coded onto the sequence are according to the sequence feature.

Figure 3 (A) Target structure of SOD2 from HKI 335 by Swiss model, (B) template SOD pdb structure (2Q2L), (C) Ramachandran plot for phi and psi bond length for model validation for SOD2 (from drought tolerent HKI 335 maize inbred) designed by PDBsum program. All r.

(D) ProSA-web used for evaluation of Z-score value of both target and template structure. (E) Superposed structure of SOD2 from ZmHKI 335 by swiss model modeling and crystal structure of putative SOD2 from Potentilla atrosanguinea by PDB. Superposition of C3 backbone of the target and the template was represented by blue and red colours. (Name of Chain_1: A22472, Name of Chain_2: B22472, Length of Chain_1: 151 residues; Length of Chain_2: 152 residues, Aligned length = 151, RMSD = 0.27; Seq ID = n_identical/n_aligned = 0.834; TM-score = 0.99714 (if normalized by length of Chain_1); TM-score = 0.99059 (if normalized by length of Chain_2); (should use TM-score normalized by length of the reference protein)).

Potentilla atrosanguinea, commonly known as Himalayan cinquefoil or ruby cinquefoil, is a vigorous herbaceous perennial of the rose family native to mountain slopes at lower elevations in the Himalayas. Surprisingly, SOD2 from Potentilla atrosanguinea have been previously reported as a highly thermostable SOD that can tolerate autoclaving as well as sub-zero temperatures (Kumar, Sahoo & Ahuja, 2002; US Patent US20070269811A1). Further, a single C95A amino acid substitution in the wild type Potentilla atrosanguinea SOD2 is known to enhance thermal properties (Kumar et al., 2012). These findings prompted us to further investigate the SOD2 from HKI 335 maize inbred.

Model validation

With the PDBsum, it has been shown that generated model of SOD2 protein revealed (~98.0% expected): 193 (84.6%) residues falling in most favoured region, (~2.0% expected): 34 (14.9%) residues in additionally allowed region, and 0.4% residues in generously allowed region with no residues in the disallowed region of the Ramachandran plot (Fig. 3C). Z-score of PROSA energy indicating overall model quality was used to check 3D models of protein structures for potential errors. In these plots displaying Z-scores, value (−6.51) of the target model was determined by X ray crystallography (represented in light blue) and nuclear magnetic resonance (represented in dark blue). This value was extremely close to the value of template 2Q2L (−6.44) (Fig. 3D). Root mean squared deviation (RMSD) value indicates the degree to which the two three dimensional structures are similar. RSMD analysis of the SOD2 model was measured from its template (2Q2l_A). The Cα RMSD and backbone RSMD deviation for both the target and the template was 0.27 Å (Fig. 3E).

Physiochemical properties

The target protein consists of very high percentages of glycine (18.5%), alanine (7.9%) and valine (9.9%) in comparison to other amino acids (Table 1). The high percentage of glycine (18.5%) shows that triple helical structure of the protein is likely to be more stable. Like that of glycine residues in protein, the proline also plays a major role in stability of helix of protein’s secondary structure. Here, the residues of proline show less percentage (5.3%). On the other hand, the percentage of glycine, valine and leucine among the hydrophobic groups are 32.56% and 17.44% respectively, while serine and threonine (both hydrophilic group) are 20.51% and 28.21% respectively. The isoelectric point (pI) value was 5.43, while the instability index was 20.1 (Table 2). The aliphatic index was 80.66, which is extremely high. The value of grand average of hydropathy (GRAVY) was −0.132 that lies between −2 and +2, suggesting that the protein is hydrophobic in nature and rated positively. The atomic composition of the protein is shown in Fig. 4.

Table 1 Amino acid composition of SOD2, Zea mays, HKI 335.

Amino acid (AA)	AA	AA (Number)	AA (%)	Hydrophobic group (%)	Hydrophilic group (%)	
Ala	A	12	7.9	13.95		
Arg	R	3	2			
Asn	N	6	4		15.38	
Asp	D	11	7.3			
Cys	C	2	1.3		5.13	
Gln	Q	3	2		7.69	
Glu	E	6	4			
Gly	G	28	18.5	32.56		
His	H	9	6		23.08	
Ile	I	8	5.3	9.3		
Leu	L	9	6	10.47		
Lys	K	6	4			
Met	M	2	1.3	2.33		
Phe	F	4	2.6	4.65		
Pro	P	8	5.3	9.3		
Ser	S	8	5.3		20.51	
Thr	T	11	7.3		28.21	
Trp	W	0	0		0	
Tyr	Y	0	0		0	
Val	V	15	9.9	17.44		
Pyl	O	0	0			
Sec	U	0	0			
	B	0	0			
	Z	0	0			
	X	0	0			

Table 2 Physio-chemical properties of SOD2.

Number of AA	Molecular weight	Theoretical pI	Formula	Negatively charged residues (Asp + Glu)	Positively charged residues (Arg + Lys)	Extinction coefficients	Instability index	Aliphatic index	GRAVY	
151	15,103.76	5.43	C650H1033N193O214S4	17	9	125	20.1	80.66	−0.132	

Figure 4 Atomic composition (i.e., number of atoms) of protein SOD2.

Post translational modification (PTM) site prediction

The post translational modification (PTM) is described as amino acid modification on the basis of protein sequence and also considered as vital issues for regulating the physiological and biological functions inside the cell. We could identify only one PTM site (phosphorylation), located at position 20 having residue S (sequence: GTDVKGTIFFSQEGDGPTTVT) with score 0.692.

Protein-protein interaction (PPI) analysis

It has been observed that the interacting partner proteins with query protein are IDP712 (uncharacterized LOC100283786 in Zea mays), sod3, pco095461 (Superoxide dismutase in Zea mays), GRMZM2G139680_P01 (2-cys peroxiredoxin BAS1 in Zea mays), SODA.3, 541646, GRMZM2G157018_P01 (ATP synthase subunit d, mitochondrial in Zea mays), GRMZM2G125151_P01 (uncharacterized LOC100283203 in Zea mays) and PER1 (period circadian regulator 1 in humans) with similarity scores 0.979, 0.883, 0.852, 0.847, 0.847, 0.837, 0.837, 0.835, 0.789 and 0.761 respectively (Fig. 5, Table 3). The process and function of SOD2 has also been compared with other proteins (Table 4). Other analysis such as protein interaction descriptions and other parameters are furnished in Supplemental Files (Tables S1 and S2).

Figure 5 Protein-protein interaction network of SOD2 with other proteins.

Table 3 Interacting proteins and their similarity scores with our query protein SOD2.

Interacting partner	Score	
IDP712	0.979	
sod3	0.883	
pco095461	0.852	
GRMZM2G139680_P01	0.847	
1E+08	0.847	
SODA.3	0.837	
541646	0.837	
GRMZM2G157018_P01	0.835	
GRMZM2G125151_P01	0.789	
PER1	0.761	

Table 4 Process and function of SOD2 compared with other proteins from protein-protein interaction.

Process	
#Term ID	Term description	Observed gene count	Background gene count	Strength	False discovery rate	
GO:0098869	Cellular oxidant detoxification	4	14	3.01	6.57E−10	
GO:0019430	Removal of superoxide radicals	3	5	3.33	6.60E−09	
GO:0006950	Response to stress	4	68	2.32	1.29E−08	
GO:0055114	Oxidation-reduction process	4	116	2.09	6.96E−08	
Functions	
GO:0016209	Antioxidant activity	4	14	3.01	1.29E−10	
GO:0004784	Superoxide dismutase activity	3	5	3.33	5.85E−09	
GO:0016491	Oxidoreductase activity	4	107	2.13	6.25E−08	
GO:0046872	Metalion binding	3	136	1.9	1.50E−05	

Domain composition prediction

Domains play a significant role in the functional activities of proteins in the cell, and are either active regularly or during the process of evolution. The domain present in SOD2 protein sequence is Sod_Cu, which is metalloprotein that helps in the prevention of damage by free radicals with the catalysation of SOD into O2 and H2O2. Two outliers homologous and homologous of known structures are also found (Table S3). The fragment GFHVHALGDTT was identified as the plant dismutase signature motif (42–52 residues). This was a consensus pattern in both target sequences of Zea mays (HKI 335) and template Potentilla atrosanguinea. GNAGGRUACGII signature motif was identified from heme ligand site which was conserved in both SOD2 of target and template sequences.

Discussion

Plants are very sensitive to number of environmental stresses such as drought, salinity and cold that can cause extreme and rapid ROS accumulation (Leng et al., 2017; Zhou et al., 2019). Plants also have important enzymatic and non-enzymatic mechanism that regulate oxidation process and help cells to protect from oxidative damage by ROS scavenging (Suzuki & Mittler, 2006). The cloning and sequencing of SOD2 from drought tolerant maize inbred line HKI 335 shows that the length of sequence is 151 amino acids with a theoretical isoelectric point of 5.43 and a predicted molecular weight of 15,103.76 kD. The utilization of BLASTp tool of NCBI we have done the homologous search of target protein with the template/reference sequences. From this analysis, best 10 homologous (reference) protein sequences on the basis of numbers of criteria like percent identity (>80%) were selected and utilized for further analysis of the protein. The analysis of amino acid sequence demonstrated that SOD_Cu were high conserved domains in the SOD2 sequence. The phylogenetic analysis of SOD2 of maize showed that orthologous sequences of SOD from different species were clustered into two clusters (i.e., cluster A and cluster B). Previous study showed that there were two Cu/ZnSODs in eukaryotes, including intracellular and extracellular Cu/ZnSODs, encoded by two different genes (Gill & Tuteja, 2010).

In secondary structure of SOD2, we found absence of alpha-helices and large number of coils and beta-sheets with fully exposed relative solvent accessibility, showing that the amino acids in the protein secondary structure are completely buried. The high value of the RSA of the target protein suggests that the protein is stable for a number of biological functions within the the cell. The Ramachandran plot from PDBsum has revealed that most of the residues falling in highly favored region and some are in additional allowed region, very less in generously allowed region with no in the disallowed region. Independently, the high percentages of these amino acids in protein regulate the synthesis of protein as well as pathways involved in signalling. Alanine with highest percentage considered as a major component of cell wall biosynthesis. Valine is involved in enhancement of glutelin, globulin, and albumin fractions in maize kernels. The high percentage of glycine shows that triple helical structure of protein is more stable. Like that of glycine residues in protein, the proline also plays a major role in stability of helix of protein’s secondary structure. Here, the sequence contains very less percentage of proline (i.e., less than 10), which indicate that the sequence is less efficient for stability of protein structure. For perfect three-dimensional structure of proteins, hydrophobic as well as hydrophilic group play important roles. Among these groups, valine and leucine (32.56% and 17.44% respectively, hydrophobic amino acids), and serine and threonine (20.51% and 28.21% respectively, hydrophilic amino acids) are majorly involved. The pI value was lesser than 7, indicate that the target protein is acidic in nature. It was estimated that the value of instability index was less than 40, which shows that the protein is stable. The value of aliphatic index was predicted to be extremely high, and therefore the target protein is considered as thermostable protein, indicating that the target protein SOD2 is resistant to decay at high temperature. This is further affirmed by comparative modeling, which led us to highly thermo-tolerant SOD2 from Potentilla atrosanguinea. The value of GRAVY lies between −2 and +2, suggesting that the target is hydrophobic in nature and rated positively. Therefore, during the cellular processes, this protein decreases the region of linking between non-polar and water molecules, and also utilize the hydrogen bonding between water molecules within the cell. There are many reports establishing that tolerance to drought and salt or drought and cold stress can be improved by exotic gene transferring from plants (Han et al., 2018; Li et al., 2019; Wu et al., 2019).

Conclusion

The Sod2 gene from drought tolerant HKI 335 maize inbred line was cloned and analyzed at protein sequence level. The HKI 335 SOD2 protein is highly similar to Potentilla atrosanguinea SOD2, which had been previously identified as highly thermostable SOD. Overall, the unique stress tolerant properties of HKI 335 SOD2, may be one of the reasons contributing to the highly drought tolerance trait exhibited by HKI 335 maize inbred line. Further work on HKI 335, especially in relation to SOD2, like in vitro characterization of the protein, overexpression, and allele mining, may lead to greater insights on the drought adaptation mechanism in maize and eventual deployment of the trait in maize hybrids.

Supplemental Information

Supplemental Information 1 Supplementary Tables.

Click here for additional data file.

Supplemental Information 2 Supplementary Figures.

Click here for additional data file.

Additional Information and Declarations

Competing Interests

Author Contributions

DNA Deposition

Data Availability

Pranjal Yadava and Sapna Langyan are Academic Editors for PeerJ.

Anuradha Gautam performed the experiments, analyzed the data, prepared figures and/or tables, authored or reviewed drafts of the article, and approved the final draft.

Fatima Nazish Khan performed the experiments, analyzed the data, prepared figures and/or tables, authored or reviewed drafts of the article, and approved the final draft.

Surabhi Priya performed the experiments, analyzed the data, prepared figures and/or tables, authored or reviewed drafts of the article, and approved the final draft.

Krishan Kumar conceived and designed the experiments, analyzed the data, authored or reviewed drafts of the article, and approved the final draft.

Shivani Sharda conceived and designed the experiments, authored or reviewed drafts of the article, and approved the final draft.

Tanushri Kaul conceived and designed the experiments, authored or reviewed drafts of the article, and approved the final draft.

Ishwar Singh conceived and designed the experiments, authored or reviewed drafts of the article, and approved the final draft.

Sapna Langyan conceived and designed the experiments, authored or reviewed drafts of the article, and approved the final draft.

Pranjal Yadava conceived and designed the experiments, analyzed the data, authored or reviewed drafts of the article, and approved the final draft.

The following information was supplied regarding the deposition of DNA sequences:

The Sod2 sequence from HKI 335 maize inbred line is available at GenBank: ALF00121.1.

The following information was supplied regarding data availability:

All necessary data has been provided in tables, figures and manuscript. Further data will be made available upon request/reasonable request.

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
