# Peer review of "Cloning and comparative modeling identifies a highly stress tolerant Cu/Zn cytosolic super oxide dismutase 2 from a drought tolerant maize inbred line"

_PeerJ, doi:10.7717/peerj.14845_

## Round 0.1 · original submission · Minor Revisions

Please modify the manuscript as per reviewer's comments

·

Basic reporting

No comment

Experimental design

no comment

Validity of the findings

no commnet

Additional comments

The manuscript entitled “Cloning and comparative modeling identifies a highly stress tolerant Cu/Zn cytosolic Super Oxide Dismutase 2 from a drought tolerant maize inbred line” is an original work that is well under the scope of the journal. The rationale of the study is well defined, and hopefully, it will provide meaningful information to the readers if published.
However, I have some comments that need to be addressed before its publication.

In the abstract, the outcomes of this study are not well mentioned. The authors should briefly mention a few important outcomes of the present work.

References are missing at various places throughout the manuscript. Authors should carefully provide references when needed throughout the manuscript. A few examples of missing references are on line number 180, 181, 274, etc.

The Authors should improve the English of the manuscript. A few suggestions are mentioned here:
1. Line 98, it should be “similar sequences”
2. Re-write the sentence “Valine is involved in……………..fractions in maize kernels.” Please see line no. 272-273.
3. Re-write the sentence “In this study,……………….protein structure.” Please see line no. 275-277.
4. Re-write the sentence “Among them…………………respectively.” Please see line no. 278-279.
5. In Line 283, the word “found” should be replaced with predicted or estimated.
6. A few sentences are redundant in the manuscript. For example, authors can delete “which means that………………at the higher temperature.” Please see line no. 284-285.

In the material and method section, the authors should provide more information. Primer sequences should also be mentioned in the material method section. Information regarding many colonies were selected for sequencing, as well as which polymerase was used (high or low fidelity), should also be provided.

The quality of images is very poor in supplementary as well as in the main manuscript.
The authors should provide well informative legends (labels) of images and tables. The legend of Figures 2 and 3 are just the same.

·

Basic reporting

The manuscript seems to be well written with professional English used throughout, however, some sentences in the manuscript needs to be rephrased as mentioned below.

1. Figure 3- “Phylogenetic analysis showing Zea mays SOD2 with other plants species SOD2 proteins from dicots and monocots” not clear what the authors want to say, rephrase the sentence.

2. Phylogenetic Analysis- Line 168- Write “The multiple sequence alignment shows aligning of SOD2 from different stress tolerant plant species” instead of “The multiple sequence alignment shows aligning of different plant species for stress tolerance”.

3. Image resolution of the figures in the manuscript (Figure 1, 2, 3, 5, 6, 7, S1, S2) is really bad and impossible to read. Please replace with better images.

Experimental design

1. The abstract does not explain why the study was done, what is the significance of the current study and what it will add to the current knowledge gap. Please add a line at the end of the abstract explaining the significance of finding similarities between the SOD2 from HKI 335 inbred maize and thermostable Potentilla atrosanguinea SOD2.

2. Introduction: Current introduction section emphasizes more on the development and breeding of drought tolerant maize varieties; however, the focus should be more on what is the mechanism behind drought tolerance. It will be even better to add some of the already known mechanisms in drought tolerant maize lines. In addition, there is no mention of thermostable Potentilla atrosanguinea SOD2 and its correlation with the current study anywhere in the introduction section.

3. The manuscript nowhere states the purpose and significance of the findings, and how the obtained knowledge from current study is going to be helpful in the future. Please rewrite and rearrange the abstract, introduction and discussion section of the manuscript accordingly.

Validity of the findings

The results in the manuscript clearly validates what the authors set to find out at the beginning which is “Cloning and comparative modeling identifies a highly stress tolerant Cu/Zn cytosolic Super Oxide Dismutase 2 from a drought tolerant maize inbred line”, however, it is not clear what was the purpose of the study and how it can be, in future, exploited to develop better drought tolerant maize varieties.

Additional comments

No Comments.

·

Basic reporting

The basic structure of the manuscript seems to be fine in accordance with the journal with well written english, explanation of the results alongwith proper figures with labelling and legends required for the understanding of the public.
I would advise if the authors could explain bit more
1)if authors could explain more regarding the role of sod as a transcriptional regulators they mentioned from the previous studies in the introduction
2) Whether sod2 also acts as a transcriptional regulator under abiotic stress?
3) What probable role it may have in the regulation of which antioxidant genes?

Experimental design

Most of the experiments are well designed to justify the aim of the study. However, I would like to suggest some minor points which the authors could incorporate in their experimental section
1) Mention the vector used for cloning SOD2
2)What were the specific conditions for growing the maize inbred line HKI 355?
3)Mention the company for the cDNA synthesis kit and nanodrop spectrophotometer used for RNA estimation
4)Mention the specific primer sequences used for colony PCR confirmation

Validity of the findings

I have a query regarding the finding of protein-protein interaction studies which the authors have shown using STRING.
Does the putative partners for protein-protein interaction of SOD2 using the STRING platform remains consistent across different species or it varies with change of the species? If the authors explain more on the robustness of the STRING platform by verifying the similar interactions using SOD2 from different species to highlight the conserved partners

Additional comments

The authors have mentioned in their title that SOD2 is Cu/Zn based enzyme. As per recently published article “C2H2 Zinc Finger Proteins Response to Abiotic Stress in Plants” by Liu et al. 2022 Sod is a C2H2 zinc finger protein and there are some conserved regions in the C2H2 zinc finger proteins. Since initially the authors have chosen SOD2 from the previous published report of the role of SOD in mitigating abiotic stress response so it would be interesting to know whether SOD2 also falls in the class of C2H2 zinc finger proteins and are the conserved motifs for C2H2 zinc finger proteins are there or they are altered for SOD2? This will also help to explain the possible role of SOD2 in transcriptional regulation.

---

## Round 0.2 · accepted · Accept

Dear Dr. Yadava,

Thank you for your submission to PeerJ.

I am writing to inform you that your manuscript - Cloning and comparative modeling identifies a highly stress tolerant Cu/Zn cytosolic Super Oxide Dismutase 2 from a drought tolerant maize inbred line - has been Accepted for publication. Congratulations!

Thanks for changing the manuscript and answering all the comments raised by the reviewers.

·

Basic reporting

NA

Experimental design

NA

Validity of the findings

NA

Additional comments

The authors have addressed almost all of the comments/questions raised earlier. I belived this revised manuscript entittled "Cloning and comparative modeling identifies a highly stress tolerant Cu/Zn cytosolic Super Oxide Dismutase 2 from a drought tolerant maize inbred line" can now be published in the journal in its present form.

·

Basic reporting

No comment

Experimental design

No comment

Validity of the findings

No comment

Additional comments

The authors have incorporated all the changes as mentioned, thereby substantially improving the quality of manuscript. I can recommend the manuscript for publication at this stage.

·

Basic reporting

No comments

Experimental design

No comments

Validity of the findings

No comments

Additional comments

The article has been revised thoroughly as per the suggestions. So, I recommend the revised article for publication.